# A CLOSER LOOK AT DEEP LEARNING HEURISTICS: LEARNING RATE RESTARTS, WARMUP AND DISTILLATION

**Akhilesh Gotmare***
Department of Computer Science
EPFL, Switzerland
akhilesh.gotmare@epfl.ch

**Nitish Shirish Keskar, Caiming Xiong & Richard Socher**
Salesforce Research
Palo Alto, US
{nkeskar, cxiong, rsocher}@salesforce.com

## ABSTRACT

The convergence rate and final performance of common deep learning models have significantly benefited from heuristics such as learning rate schedules, knowledge distillation, skip connections, and normalization layers. In the absence of theoretical underpinnings, controlled experiments aimed at explaining these strategies can aid our understanding of deep learning landscapes and the training dynamics. Existing approaches for empirical analysis rely on tools of linear interpolation and visualizations with dimensionality reduction, each with their limitations. Instead, we revisit such analysis of heuristics through the lens of recently proposed methods for loss surface and representation analysis, viz., mode connectivity and canonical correlation analysis (CCA), and hypothesize reasons for the success of the heuristics. In particular, we explore knowledge distillation and learning rate heuristics of (cosine) restarts and warmup using mode connectivity and CCA. Our empirical analysis suggests that: (a) the reasons often quoted for the success of cosine annealing are not evidenced in practice; (b) that the effect of learning rate warmup is to prevent the deeper layers from creating training instability; and (c) that the latent knowledge shared by the teacher is primarily disbursed to the deeper layers.

## 1 INTRODUCTION

The introduction of heuristics such as normalization layers (Ioffe & Szegedy, 2015; Ba et al., 2016), residual connections (He et al., 2016), and learning rate strategies (Loshchilov & Hutter, 2016; Goyal et al., 2017; Smith, 2017) have greatly accelerated progress in Deep Learning. Many of these ingredients are now commonplace in modern architectures, and some of them have also been buttressed with theoretical guarantees (Balduzzi et al., 2017; Poggio & Liao, 2017; Hardt & Ma, 2016). However, despite their simplicity and efficacy, why some of these heuristics work is still relatively unknown. Existing attempts at explaining these strategies empirically have been limited to intuitive explanations and the use of tools such as spectrum analysis (Sagun et al., 2017), linear interpolation between two models and low-dimensional visualizations (Li et al., 2017) of the loss surface. In our work, we instead use recent tools built specifically for analyzing deep networks, viz., mode connectivity (Garipov et al., 2018) and singular value canonical correlation analysis (SVCCA) (Raghu et al., 2017). We investigate three strategies in detail: (a) cosine learning rate decay, (b) learning rate warmup, and (c) knowledge distillation, and list the summary of our contributions at the end of this section.

Cosine annealing (Loshchilov & Hutter, 2016), also known as stochastic gradient descent with restarts (SGDR), and more generally cyclical learning rate strategies (Smith, 2017), have been recently proposed to accelerate training of deep networks (Coleman et al., 2018). The strategy involves reductions and restarts of learning rates over the course of training, and was motivated as means to escape spurious local minima. Experimental results have shown that SGDR often improves convergence both from the standpoint of iterations needed for convergence and the final objective.

---

*Work performed while interning at Salesforce Research

Learning rate warmup (Goyal et al., 2017) also constitutes an important ingredient in training deep networks, especially in the presence of large or dynamic batch sizes. It involves increasing the learning rate to a large value over a certain number of training iterations followed by decreasing the learning rate, which can be performed using step-decay, exponential decay or other such schemes. The strategy was proposed out of the need to induce stability in the initial phase of training with large learning rates (due to large batch sizes). It has been employed in training of several architectures at scale including ResNets and Transformer networks (Vaswani et al., 2017).

Further, we investigate knowledge distillation (KD) (Hinton et al., 2015). This strategy involves first training a (teacher) model on a typical loss function on the available data. Next, a different (student) model (typically much smaller than the teacher model) is trained, but instead of optimizing the loss function defined using hard data labels, this student model is trained to mimic the teacher model. It has been empirically found that a student network trained in this fashion significantly outperforms an identical network trained with the hard data labels. We defer a detailed discussion of the three heuristics, and existing explanations for their efficacy to sections 3, 4 and 5 respectively.

Finally, we briefly describe the tools we employ for analyzing the aforementioned heuristics. Mode connectivity (MC) is a recent observation that shows that, under circumstances, it is possible to connect any two local minima of deep networks via a piecewise-linear curve (Garipov et al., 2018; Draxler et al., 2018). This shows that local optima obtained through different means, and exhibiting different local and generalization properties, are connected. The authors propose an algorithm that locates such a curve. While not proposed as such, we employ this framework to better understand loss surfaces but begin our analysis in Section 2 by first establishing its robustness as a framework.

Deep network analyses focusing on the *weights* of a network are inherently limited since there are several invariances in this, such as permutation and scaling. Recently, Raghu et al. (2017) propose using CCA along with some pre-processing steps to analyze the *activations* of networks, such that the resulting comparison is not dependent on permutations and scaling of neurons. They also prove the computational gains of using CCA over alternatives ((Li et al., 2015)) for representational analysis and employ it to better understand many phenomenon in deep learning.

**Contributions:**

- We use mode connectivity and CCA to improve understanding of cosine annealing, learning rate warmup and knowledge distillation. For mode connectivity, we also establish the robustness of the approach across changes in training choices for obtaining the modes.
- We demonstrate that the reasons often quoted for the success of cosine annealing are not substantiated by our experiments, and that the iterates move over barriers after restarts but the explanation of escaping local minima might be an oversimplification.
- We show that learning rate warmup primarily limits weight changes in the deeper layers and that freezing them achieves similar outcomes as warmup.
- We show that the latent knowledge shared by the teacher in knowledge distillation is primarily disbursed in the deeper layers.

## 2 EMPIRICAL TOOLS

### 2.1 MODE CONNECTIVITY

Garipov et al. (2018) introduce a framework, called mode connectivity, to obtain a low loss (or high accuracy, in the case of classification) curve of simple form, such as a piecewise linear curve, that connects optima (modes of the loss function) found independently. This observation suggests that points at the same loss function depth are connected, somewhat contrary to several empirical results claiming that minima are isolated or have barriers between them[1].

Let $w_a \in \mathbb{R}^D$ and $w_b \in \mathbb{R}^D$ be two modes in the $D$-dimensional parameter space obtained by optimizing a given loss function $\mathcal{L}(w)$ (like the cross-entropy loss). We represent a curve connecting

---

[1]Draxler et al. (2018) independently report the same observation for neural network loss landscapes, and claim that this is suggestive of the resilience of neural networks to perturbations in model parameters.

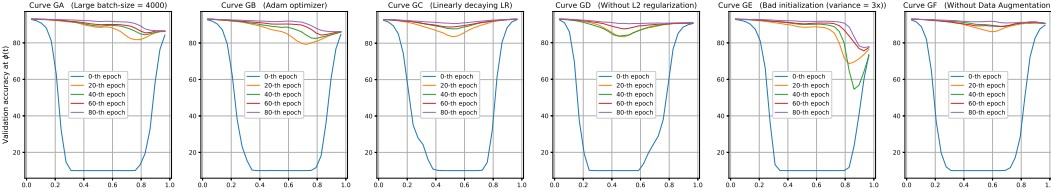

Figure 1: Validation accuracy corresponding to models on the following 6 different curves - curve $GA$ represents curve connecting mode $G$ (one found with default hyperparameters) and mode $A$ (using large batch size), similarly, curve $GB$ connects mode $G$ and mode $B$ (using Adam), curve $GC$ connects to mode $C$ (using linearly decaying learning rate), curve $GD$ to mode $D$ (with lesser L2 regularization), curve $GE$ to mode $E$ (using a poor initialization), and curve $GF$ to mode $F$ (without using data augmentation). $t = 0$ corresponds to mode $G$ for all plots.

$w_a$ and $w_b$ by $\phi_\theta(t) : [0,1] \rightarrow \mathbb{R}^D$, such that $\phi_\theta(0) = w_a$ and $\phi_\theta(1) = w_b$. To find a low loss path, we find the set of parameters $\theta \in \mathbb{R}^D$ that minimizes the following loss: $\ell(\theta) = \int_0^1 \mathcal{L}(\phi_\theta(t))dt = \mathbb{E}_{t \sim U(0,1)} \mathcal{L}(\phi_\theta(t))$ where $U(0,1)$ is the uniform distribution in the interval $[0,1]$. To optimize $\ell(\theta)$ for $\theta$, we first need to chose a parametric form for $\phi_\theta(t)$. One of the forms proposed by Garipov et al. (2018) is a polygonal chain with a single bend at $\theta$ as follows

$$\phi_\theta(t) = \begin{cases} 2(t\theta + (0.5 - t)w_a), & \text{if } 0 \leq t \leq 0.5 \\ 2((t - 0.5)w_b + (1 - t)\theta) & \text{if } 0.5 < t \leq 1 \end{cases}$$

To minimize $\ell(\theta)$, we sample $t \sim U[0,1]$ at each iteration and use $\nabla_\theta \mathcal{L}(\phi_\theta(t))$ as an unbiased estimate for the true gradient $\nabla_\theta \ell(\theta)$ to perform updates on $\theta$, where $\theta$ is initialized with $\frac{1}{2}(w_a + w_b)$.

### 2.1.1 RESILIENCE OF MODE CONNECTIVITY

To demonstrate that the curve-finding approach works in practice, Garipov et al. (2018) use two optima found using different initializations but a common training scheme which we detail below. We explore the limits of this procedure by connecting optima obtained from different training strategies. Our goal of this investigation is to first establish the robustness of the framework in order to seamlessly use it as a tool for analysis. In particular, we experiment with different initializations, optimizers, data augmentation choices, and hyperparameter settings including regularization, training batch sizes, and learning rate schemes. We note in passing that while the framework was proposed to connect two points in the parameter space that are at equal depth in the loss landscape, it is well-defined to also connect points at different depths; in this case, the path corresponds to one that minimizes the average loss along the curve.

Conventional wisdom suggests that the different training schemes mentioned above will converge to regions in the parameter space that are vastly different from each other. Examples of this include size of minibatches used during training (Keskar et al., 2016), choice of optimizer (Heusel et al., 2017; Wilson et al., 2017), initialization (Goodfellow et al., 2016) and choice of regularizer. Having a high accuracy connection between these pairs would seem counterintuitive.

For obtaining the reference model (named mode $G$), we train the VGG-16 model architecture (Simonyan & Zisserman, 2014) using CIFAR-10 training data (Krizhevsky et al., 2014) for 200 epochs with SGD. We then build 6 variants of the reference mode $G$ as follows: we obtain mode $A$ using a training batch size of 4000, mode $B$ by using the Adam optimizer instead of SGD, mode $C$ with a linearly decaying learning rate instead of the step decay used in mode $G$, mode $D$ using a smaller weight decay of $5 \times 10^{-6}$, mode $E$ by increasing the variance of the initialization distribution to $3 \times \sqrt{2/n}$ and mode $F$ using no data augmentation. Note that for the set of modes $\{A, B, C, D, E, F\}$, all the other hyper-parameters and settings except the ones mentioned above are kept same as that for mode $G$. We use the mode connectivity algorithm on each of the 6 pairs of modes including $G$ and another mode, resulting in curves $GA$, $GB$, $GC$, $GD$, $GE$, and $GF$.

Figure 1 shows the validation accuracy for models on each of the 6 connecting curves during the 20th, 40th, 60th and 80th epochs of the mode connectivity training procedure and also for models on the line segment joining the two endpoints (corresponding to the initialization for $\theta$ at epoch

0). As described in Section 2.1, for a polychain curve $GX$ (connecting modes $G$ and $X$ using the curve described by $\theta$), model parameters $\phi_\theta(t)$ on the curve are given by $p_{\phi_\theta(t)} = 2(tp_\theta + (0.5 - t)p_G)$ if $0 \leq t \leq 0.5$ and $p_{\phi_\theta(t)} = 2((t - 0.5)p_X + (1 - t)p_\theta)$ if $0.5 < t \leq 1$ where $p_G$, $p_\theta$ and $p_X$ are parameters of the models $G$, $\theta$, and $X$ respectively. Thus $\phi_\theta(0) = G$ and $\phi_\theta(1) = X$.

In a few epochs of the curve training, for all 6 pairs, we can find a curve such that each point on it generalizes almost as well as models from the pair that is being connected. Note that by virtue of existence of these 6 curves, there exists a high accuracy connecting curve (albeit with multiple bends) for each of the $\binom{7}{2}$ pairs of modes. We refer the reader to Appendix 7 for a t-SNE plot of the modes and their connections, and also for additional plots and details. Having established the high likelihood of the existence of these curves, we use this procedure along with interpolation of the loss surface between parameters at different epochs as tools to analyze the dynamics of SGD and SGDR.

## 2.2 CCA FOR MEASURING REPRESENTATIONAL SIMILARITY

Canonical correlation analysis (CCA) is a classical tool from multivariate statistics (Hotelling, 1936) that investigates the relationships between two sets of random variables. Raghu et al. (2017) have proposed coupling CCA with pre-processing steps like Singular Value Decomposition (SVD) or Discrete Fourier Transform (DFT) to design a similarity metric for two neural net layers that we want to compare. These layers do not have to be of the same size or belong to the same network.

Given a dataset with $m$ examples $X = \{x_1, \ldots x_m\}$, we denote the scalar output of the neuron $z_i^l$ ($i$-th neuron of layer $l$) for the input $x_i$ by $f_{z_i^L}(x_i)$. These scalar outputs can be stacked (along $n$ different neurons and $m$ different datapoints) to create a matrix $L \in \mathbb{R}^{m \times n}$ representing the output of a layer corresponding to the entire dataset. This choice of comparing neural network layers using activations instead of weights and biases is crucial to the setup proposed. Indeed, invariances due to re-parameterizations and permutations limit the interpretability of the model weights (Dinh et al., 2017). However, under CCA of the layers, two activation sets are comparable by design.

Given representations corresponding to two layers $L_a \in \mathbb{R}^{m_a \times n}$ and $L_b \in \mathbb{R}^{m_b \times n}$, SVCCA first performs dimensionality reduction using SVD to obtain $L_a' \in \mathbb{R}^{m_a' \times n}$ and $L_b' \in \mathbb{R}^{m_b' \times n}$ while preserving 99% of the variance. The subsequent CCA step involves transforming $L_a'$ and $L_b'$ to $a_1^\top L_a'$ and $b_1^\top L_b'$ respectively where $\{a_1, b_1\}$ is found by maximizing the correlation between the transformed subspaces, and the corresponding correlation is denoted by $\rho_1$. This process continues, using orthogonality constraints, till $c = \min\{m_a', m_b'\}$ leading to the set of correlation values $\{\rho_1, \rho_2 \ldots \rho_c\}$ corresponding to $c$ pairs of canonical variables $\{\{a_1, b_1\}, \{a_2, b_2\}, \ldots \{a_c, b_c\}\}$ respectively. We refer the reader to Raghu et al. (2017) for details on solving these optimization problems. The average of these $c$ correlations $\frac{1}{n} \sum_i \rho_i$ is then considered as a measure of the similarity between the two layers. For convolutional layers, Raghu et al. (2017) suggest using a DFT pre-processing step before CCA, since they typically have a large number of neurons ($m_a$ or $m_b$), where performing raw SVD and CCA would be computationally too expensive. This procedure can then be employed to compare different neural network representations and to determine how representations evolve over training iterations.

## 3 STOCHASTIC GRADIENT DESCENT WITH RESTARTS (SGDR)

Loshchilov & Hutter (2016) introduced SGDR as a modification to the common linear or step-wise decay of learning rates. The strategy decays learning rates along a cosine curve and then, at the end of the decay, restarts them to its initial value. The learning rate at the $t$-th epoch in SGDR is given by the following expression in (1) where $\eta_{min}$ and $\eta_{max}$ are the lower and upper bounds respectively for the learning rate. $T_{cur}$ represents how many epochs have been performed since the last restart and a warm restart is simulated once $T_i$ epochs are performed. Also $T_i = T_{mult} \times T_{i-1}$, meaning the period $T_i$ for the learning rate variation is increased by a factor of $T_{mult}$ after each restart.

$$\eta_t = \eta_{min} + \frac{1}{2}(\eta_{max} - \eta_{min})\left(1 + \cos\left(\frac{T_{cur}}{T_i}\pi\right)\right) \tag{1}$$

While the strategy has been claimed to outperform other learning rate schedulers, little is known why this has been the case. One explanation that has been given in support of SGDR is that it

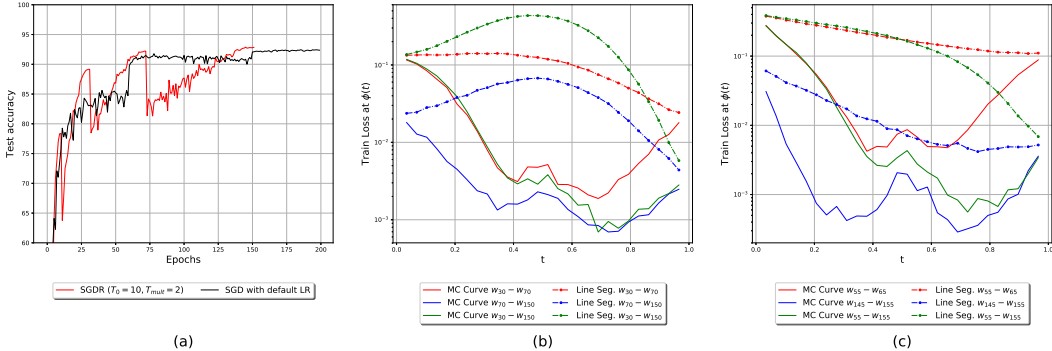

Figure 2: (a) Validation accuracy of a VGG16 model trained on CIFAR-10 using SGDR with warm restarts simulated every $T_0 = 10$ epochs and $T_{mult} = 2$. (b) Cross-entropy training loss on the curve found through Mode Connectivity (MC Curve) and on the line segment (Line Seg.) joining modes $w_{30}$ (model corresponding to parameters at the 30-th epoch of **SGDR**) and $w_{70}$, $w_{70}$ and $w_{150}$, $w_{30}$ and $w_{150}$. (c) Cross-entropy training loss on the curve found through Mode Connectivity (MC Curve) and on the line segment (Line Seg.) joining modes $w_{55}$ (model corresponding to parameters at the 55-th epoch of **SGD with step decay learning rate scheme**) and $w_{65}$, $w_{145}$ and $w_{155}$, $w_{55}$ and $w_{155}$.

can be useful to deal with multi-modal functions, where the iterates could get stuck in a local optimum and a restart will help them get out of it and explore another region; however, Loshchilov & Hutter (2016) do not claim to observe any effect related to multi-modality. Huang et al. (2017) propose an ensembling strategy using the set of iterates before restarts and claim that, when using the learning rate annealing cycles, the optimization path converges to and escapes from several local minima. We empirically investigate if this is actually the case by interpolating the loss surface between parameters at different epochs and studying the training and validation loss for parameters on the hyperplane passing through[2] the two modes found by SGDR and their connectivity. Further, by employing the CCA framework as described in Section 2.2, we investigate the progression of training, and the effect of restarts on the model activations.

We train a VGG-16 network (Simonyan & Zisserman, 2014) on the CIFAR-10 dataset using SGDR. For our experiments, we choose $T_0 = 10$ epochs and $T_{mult} = 2$ (warm restarts simulated every 10 epochs and the period $T_i$ doubled at every new warm restart), $\eta_{max} = 0.05$ and $\eta_{min} = 10^{-6}$. We also perform VGG training using SGD (with momentum of 0.9) and a step decay learning rate scheme (initial learning rate of $\eta_0 = 0.05$, scaled by 5 at epochs 60 and 150). Figure 2(a) shows the validation accuracy over training epochs with these two learning rate schemes.

In order to understand the loss landscape on the optimization path of SGDR, the pairs of iterates obtained just before the restarts $\{w_{30}, w_{70}\}$, $\{w_{70}, w_{150}\}$ and $\{w_{30}, w_{150}\}$ are given as inputs to the mode connectivity algorithm, where $w_n$ is the model corresponding to parameters at the $n$-th epoch of training. Figure 2(b) shows the training loss for models along the line segment joining these pairs and those on the curve found through mode connectivity. For the baseline case of SGD training, we connect the iterates around the epochs when we decrease our learning rate in the step decay learning rate scheme. Thus, we chose $\{w_{55}, w_{65}\}$, $\{w_{145}, w_{165}\}$ and $\{w_{55}, w_{165}\}$ as input pairs to the mode connectivity algorithm. Figure 2(c) shows the training loss for models

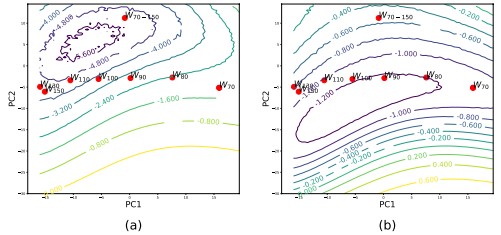

Figure 3: (a) Training loss surface and (b) validation loss surface, log scales, for points on the plane defined by $\{w_{70}, w_{150}, w_{70-150}\}$ including projections of the SGDR iterates on this hyperplane. A curve of a given color represents a contour line, with the log-loss (lower being better) corresponding to this contour shown in the same color.

---

[2]This hyperplane is the set of all affine combinations of $w_a$, $w_b$ and their connection $\theta = w_{a-b}$.

along the line segments joining these pairs and
the curves found through mode connectivity.

From Figure 2(b), it is clear that for the pairs $\{w_{30}, w_{150}\}$ and $\{w_{70}, w_{150}\}$ the training loss for points on segment is much higher than the endpoints suggesting that SGDR indeed finds paths that move over a barrier[3] in the training loss landscape. In contrast, for SGD (without restarts) in Figure 2(c) none of the three pairs show evidence of having a training loss barrier on the line segment joining them. Instead there seems to be an almost linear decrease of training loss along the direction of these line segments, suggesting that SGD's trajectory is quite different from SGDR's. We present additional experiments, including results for other metrics, in Appendix 8.

To further understand the SGDR trajectory, we evaluate the intermediate points on the hyperplane in the $D$-dimensional space defined by the three points: $w_{70}$, $w_{150}$ and $w_{70-150}$, where $w_{70-150}$ is the bend point that defines the high accuracy connection for the pair $\{w_{70}, w_{150}\}$. Figures 3(a) and 3(b) show the training and validation loss surface for points in this subspace, respectively. Note that the intermediate iterates do not necessarily lie in this plane, and thus are projected. We refer the reader to Appendix 8 for additional details on the projection, and analogous results with $w_{30}$ and $w_{70}$. Results for the VGG-16 architecture with batch-normalization are also presented in Appendix 8.4.

Figure 3(a) suggests that SGDR helps the iterates converge to a different region although neither of $w_{70}$ or $w_{150}$ are technically a local minimum, nor do they appear to be lying in different *basins*, hinting that Huang et al. (2017)'s claims about SGDR converging to and escaping from local minima might be an oversimplification.[4] Another insight we can draw from Figure 3(a) is that the path found by mode connectivity corresponds to lower training loss than the loss at the iterates that SGDR converges to ($\mathcal{L}(w_{150}) > \mathcal{L}(w_{70-150})$). However, Figure 3(b) shows that models on this curve seem to overfit and not generalize as well as the iterates $w_{70}$ and $w_{150}$. Thus, although gathering models from this connecting curve might seem as a novel and computationally cheap way of creating ensembles, this generalization gap alludes to one limitation in doing so; Garipov et al. (2018) point to other shortcomings of curve ensembling in their original work. In Figure 3, the region of the plane between the iterates $w_{70}$ and $w_{150}$ corresponds to higher training loss but lower validation loss than the two iterates. This hints at a reason why averaging iterates to improve generalization using cyclic or constant learning rates (Izmailov et al., 2018) has been found to work well.

Finally, in Figure 14 in Appendix 9, we present the CCA similarity plots for two pairs of models: epochs 10 and 150 (model at the beginning and end of training), and epochs 150 and 155 (model just before and just after a restart). For standard SGD training, Raghu et al. (2017) observe that the activations of the shallower layers bear closer resemblance than the deeper layers between a partially and fully trained network from a given training run. For SGDR training, we witness similar results (discussed in Appendix 9), meaning that the representational similarities between the network layers at the beginning and end of training are alike for SGDR and SGD, even though restarts lead to a trajectory that tends to cross over barriers.

## 4 WARMUP LEARNING RATE SCHEME

Learning rate warmup is a common heuristic used by many practitioners for training deep neural nets for computer vision (Goyal et al., 2017) and natural language processing (Bogoychev et al., 2018; Vaswani et al., 2017) tasks. Theoretically, it can be shown that the learning dynamics of SGD rely on the ratio of the batch size and learning rate (Smith et al., 2017; Jastrzebski et al., 2017; Hoffer et al., 2017). And hence, an increase in batch size over a baseline requires an accompanying increase in learning rate for comparable training. However, in cases when the batch size is increased significantly, the curvature of the loss function typically does not support a proportional increase in the learning rate. Warmup is hence motivated as a means to use large learning rates without causing training instability. We particularly focus on the importance of the learning rate schedule's warmup phase in the large batch (LB) training of deep convolutional neural networks as discussed in Goyal

---

[3]a path is said to have moved over or crossed a barrier between epoch $m$ and $n$ ($n > m$) if $\exists\ w_t \in \{\lambda w_m + (1 - \lambda)w_n | \lambda \in [0, 1]\}$ such that $\mathcal{L}(w_t) > \max\{\mathcal{L}(w_m), \mathcal{L}(w_n)\}$

[4]We note in passing that during our experiments, a strategy that consistently performed well is one of a cosine (or linear) decay over the entire budget. We hypothesize that this decay, and less so the restarts, plays a major role in the success of such strategies.

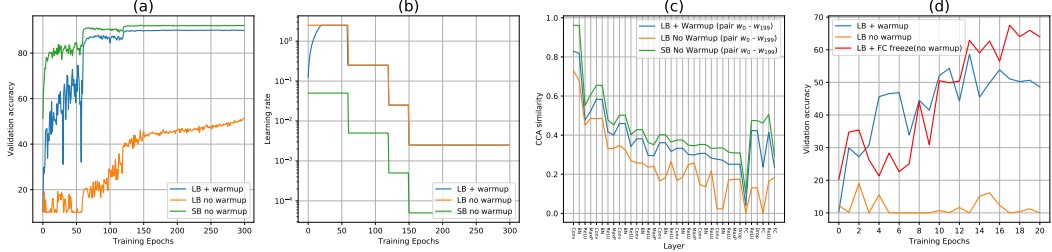

Figure 4: (a) Validation accuracy and (b) Learning rate for the three training setups (c) CCA similarity for $i$-th layer from two different iterations (0-th (before warmup) and 200-th (after warmup) during training (d) Comparing warmup and FC freezing strategies on VGG11 training

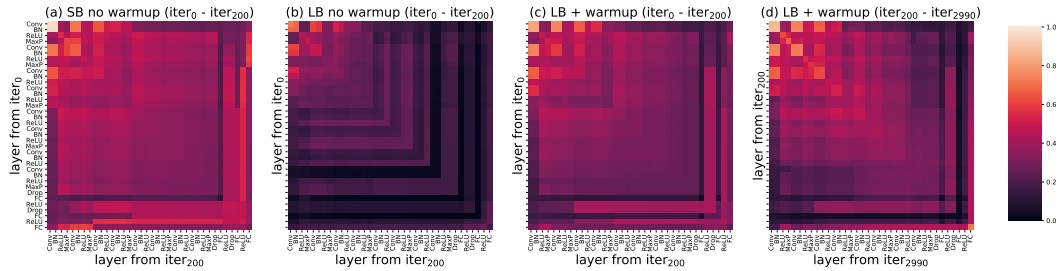

Figure 5: CCA similarity output plots for (**a**) SB no warmup, (**b**) LB no warmup, (**c, d**) LB + warmup training. The $i, j$-th cell represents the CCA similarity between layer $i$ of the first model, and layer $j$ of other. A higher score implies that the layers are more similar (lighter color).

et al. (2017). Their work adopts a linear scaling rule for adjusting the learning rate as a function of the minibatch size, to enable large-batch training. The question we aim to investigate here is: *How does learning rate warmup impact different layers of the network?*

Using CCA as a tool to study the learning dynamics of neural networks through training iterations, we investigate the differences and similarities for the following 3 training configurations - (a) large batch training with warmup (LB + warmup), (b) large batch training without warmup (LB no warmup) and (c) small batch training without warmup (SB no warmup). We train a VGG-11 architecture on the CIFAR-10 (Krizhevsky et al., 2014) dataset using SGD with momentum of 0.9. Learning rate for the small batch case (batch-size of 100) is set to 0.05, and for the large batch cases (batch-size of 5000) is set to 2.5 as per the scaling rule. For the warmup, we increase the learning rate from 0 to 2.5 over the first 200 iterations. Subsequently, we decrease the learning rate as per the step decay schedule for all runs, scaling it down by a factor of 10 at epochs 60, 120 and 150. We plot the learning rate and validation accuracy for these 3 cases in Figure 4(b) and (a).

Using CCA and denoting the model at the $j$-th iteration of a training setup by $iter_j$, we compare activation layers from $iter_0$ (init.) and $iter_{200}$ (end of warmup) for each of the three runs, presented in Figures 5(a), (b) and (c), and also layers from $iter_{200}$ (end of warmup) and $iter_{2990}$ (end of training) for the LB + warmup case, presented in Figure 5(d). Figure 4(c) plots the similarity for layer $i$ of $iter_a$ with the same layer of $iter_b$ (this corresponds to diagonal elements of the matrices in Figure 5) for these three setups.

An evident pattern in Figures 5(a), (b) and (c) is the increase in similarity for the last few layers (stack of fully-connected layers) for the LB + warmup and SB cases, which is absent in the LB without warmup case. This suggests that when used with the large batch size and learning rate, warmup tends to avoid unstably large changes in the fully-connected (FC) stack for this network configuration. To validate this proposition, we train using the LB without warmup setup, but freezing the fully-connected stack for the first 20 epochs[5] (LB no warmup + FC freeze). Figure 4(d) shows the validation accuracy for this training run in comparison to the three training setups discussed before. The performance is comparable at the end of warmup by freezing the FC stack,

---

[5]equivalent to 200 iterations for large batch-size of 5k (CIFAR-10 training dataset size is 50k)

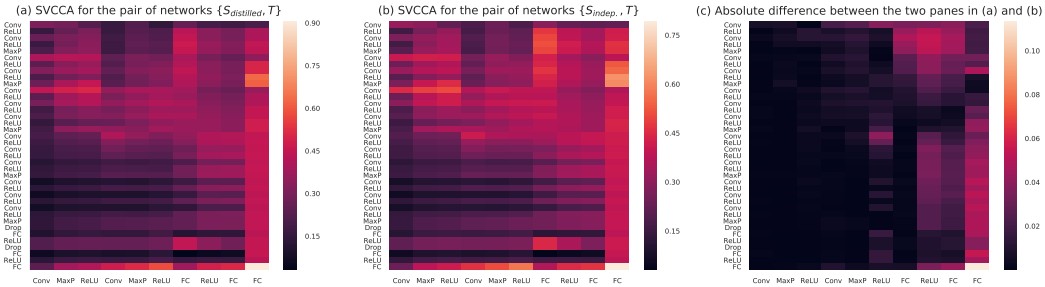

Figure 6: CCA similarity between $S_{\text{distilled}}$ - $T$, $S_{\text{indep.}}$ - $T$, and their difference. $i, j$-th cell of the difference plot represents $|\text{CCA}(l_T^i, l_{S_{\text{distilled}}}^j) - \text{CCA}(l_T^i, l_{S_{\text{indep.}}}^j)|$ where $l_M^i$ denotes the $i$-th layer of network $M$, $T$ denotes the teacher network (VGG16), $S_{\text{distilled}}$ is the student network trained using distillation and $S_{\text{indep.}}$ is the student network trained using hard training labels.

suggesting the validity our proposition in this case. We refer the reader to Appendix 10 for analogous results for ResNet-18 and ResNet-32 (He et al., 2016); thus also demonstrating the generality of our claim. Finally, note from Figure 4(d) that no qualitative difference exists in the trajectory beyond the warmup when compared to the standard training approach (Raghu et al., 2017).

## 5    KNOWLEDGE DISTILLATION

We study knowledge distillation as proposed by Hinton et al. (2015) using CCA to measure representational similarity between layers of the teacher and student model. Distillation involves training a "student" model using the output probability distribution of a "teacher" model. This has been widely known to help the student model perform better than it would, if it were trained using hard labels due to knowledge transfer from the teacher model. The reason often quoted for the success of distillation is the transfer of *dark knowledge* from the teacher to the student (Hinton et al., 2015), and more recently, as an interpretation of importance weighing (Furlanello et al., 2018). We investigate if this knowledge transfer is limited to certain parts of the network, and if representational similarity between layers of the student and teacher model and a student can help answer this question.

To construct an example of distillation that can be used for our analysis, we use a VGG-16 model (Simonyan & Zisserman, 2014) as our teacher network and a shallow convolutional network (`[conv, maxpool, relu]x2, fc, relu, fc, fc, softmax`) as the student network. We train the shallow network for CIFAR-10 using the teacher's predicted probability distribution (softened using a temperature of 5), ($S_{\text{distilled}}$), and for the baseline, train another instance of the same model in a standard way using hard labels, ($S_{\text{indep.}}$). Over 5 runs for each of the two setups, we find the distillation training attains the best validation accuracy at $85.18\%$ while standard training attains its best at $83.01\%$. We compare their layer-wise representations with those of the teacher network ($T$).

Figure 6 shows the CCA plots and the absolute value of their difference. The scores of these two pairs are quite similar for the shallow layers of the student network relative to the deeper layers, suggesting that the difference that knowledge distillation brings to the training of smaller networks is restricted to the deeper layers (`fc` stack). Similar results are obtained through different configurations for the student and teacher when the student benefits from the teacher's knowledge. We hypothesize that the *dark* knowledge transferred by the teacher is localized majorly in the deeper (discriminative) layers, and less so in the feature extraction layers. We also note that this is not dissimilar to the hypothesis of Furlanello et al. (2018), and also relates ot the results from the literature on fine-tuning or transfer learning (Goodfellow et al., 2016; Yosinski et al., 2014; Howard & Ruder, 2018) which suggest training of only higher layers.

## 6    DISCUSSION AND CONCLUSION

Heuristics have played an important role in accelerating progress of deep learning. Founded in empirical experience, intuition and observations, many of these strategies are now commonplace

in architectures. In the absence of strong theoretical guarantees, controlled experiments aimed at explaining the the efficacy of these strategies can aid our understanding of deep learning and the training dynamics. The primary goal of our work was the investigation of three such heuristics using sophisticated tools for landscape analysis. Specifically, we investigate cosine annealing, learning rate warmup, and knowledge distillation. For this purpose, we employ recently proposed tools of mode connectivity and CCA. Our empirical analysis sheds light on these heuristics and suggests that: (a) the reasons often quoted for the success of cosine annealing are not evidenced in practice; (b) that the effect of learning rate warmup is to prevent the deeper layers from creating training instability; and (c) that the latent knowledge shared by the teacher is primarily disbursed in the deeper layers.

Inadvertently, our investigation also leads to the design of new heuristics for practically improving the training process. Through our results on SGDR, we provide additional evidence for the success of averaging schemes in this context. Given the empirical results suggesting the localization of the knowledge transfer between teacher and student in the process of distillation, a heuristic can be designed that only trains portions of the (pre-trained) student networks instead of the whole network. For instance, recent results on self-distillation (Furlanello et al., 2018) show improved performance via multiple generations of knowledge distillation for the same model. Given our results, computational costs of subsequent generations can be reduced if only subsets of the model are trained, instead of training the entire model. Finally, the freezing of weights instead of employing learning rate warmup allows for comparable training performance but with reduced computation during the warmup phase. We note in passing that our result also ties in with results of Hoffer et al. (2018) who suggest not training the classifier *at all* with negligible loss in performance. Our empirical experiments and hypotheses open new questions and encourage a deeper exploration into improving and better understanding these heuristics.

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

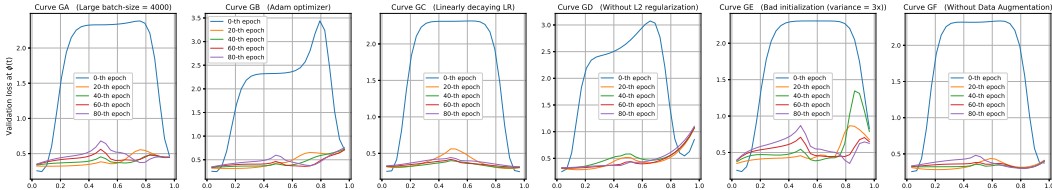

Figure 7: Validation loss corresponding to models on the 6 different curves

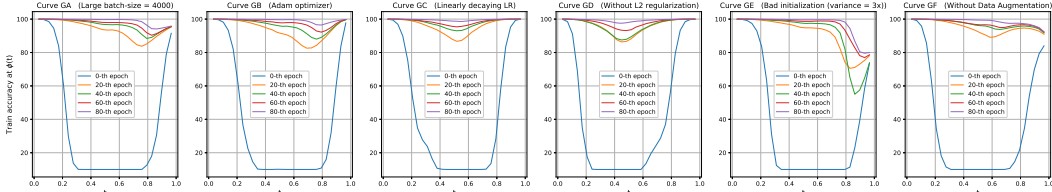

Figure 8: Training accuracy corresponding to models on the 6 different curves

## APPENDIX

## 7 ADDITIONAL RESULTS ON ROBUSTNESS OF MC

### 7.1 TRAINING DETAILS

The learning rate is initialized to $0.05$ and scaled down by a factor of $5$ at epochs $\{60, 120, 160\}$ (step decay). We use a training batch size of $100$, momentum of $0.9$, and a weight decay of $0.0005$. Elements of the weight vector corresponding to a neuron are initialized randomly from the normal distribution $\mathcal{N}(0, \sqrt{2/n})$ where $n$ is the number of inputs to the neuron. We also use data augmentation by random cropping of input images.

### 7.2 PLOTS

Figures 7, 8 and 9 show the Validation Loss, Training Accuracy and Training Loss respectively for the curves joining the 6 pairs discussed in Section 2.1.1. These results too, confirm the overfitting or poor generalization tendency of models on the curve.

### 7.3 T-SNE VISUALIZATION FOR THE 7 MODES

We use t-SNE (Maaten & Hinton, 2008) to visualize these 7 modes and the $\theta$ points that define the connectivity for the 6 pairs presented in Section 2.1.1, in a 2-dimensional plot in Figure 10. Since t-SNE is known to map only local information correctly and not preserve global distances, we caution the reader about the limited interpretability of this visualization, it is presented simply to establish the notion of connected modes.

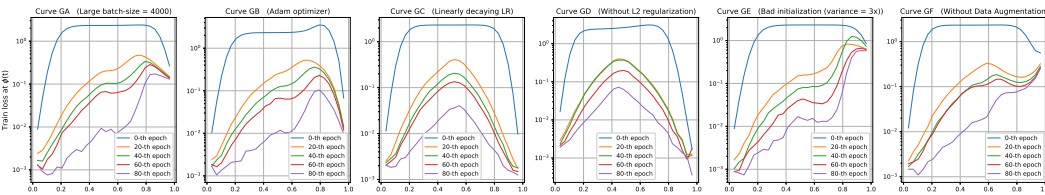

Figure 9: Training loss corresponding to models on the 6 different curves.

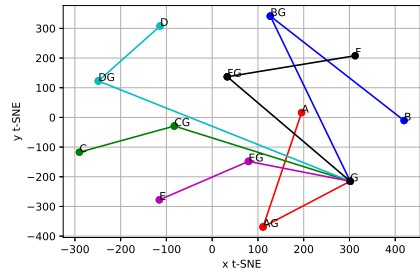

Figure 10: Representing the modes and their connecting point using t-SNE

# 8 ADDITIONAL SGDR RESULTS

## 8.1 ADDITIONAL RESULTS

For completeness, in Figure 11, we present the Validation loss, Validation accuracy and Training accuracy results for the curves and line segments joining iterates from SGDR and SGDR discussed in Figure 2(c) and (d).

## 8.2 PROJECTING ITERATES

The $W_n$ in Figure 3 is equivalent to

$$W_n = P_c(w_n) = \lambda^{\star\top} \begin{bmatrix} w_{70} \\ w_{150} \\ \theta \end{bmatrix}$$

$$\text{where } \lambda^{\star} = \text{argmin}_{\lambda \in \mathbb{R}^3} ||\lambda^{\top} \begin{bmatrix} w_{70} \\ w_{150} \\ \theta \end{bmatrix} - w_n||_2^2$$

meaning it is the point on the plane (linear combination of $w_{70}, w_{150}$ and $\theta$) with the least $l$-2 distance from the original point (iterate in this case).

## 8.3 CONNECTING MODES $w_{30}$ AND $w_{70}$ FROM SGDR

In Section 3, we present some experiments and make observations on the trajectory of SGDR by using the plane defined by the points $w_{70}$, $w_{150}$ and $w_{70-150}$. Here we plot the Training loss and Validation loss surface in Figure 12 for another plane defined by SGDR's iterates $w_{30}, w_{70}$ and their connection $w_{30-70}$ to ensure the reader that the observations made are general enough.

## 8.4 RESULTS FOR VGG-16 WITH BATCH NORMALIZATION

The VGG-16 architecture used in Section 3 does not include Batch Normalization, which has been known to alter properties of the loss surface (Santurkar et al. (2018)). Therefore we train VGG-16 with Batch Normalization using SGDR to verify if our observations hold for this case too. As pointed out in Appendix A.2 of Garipov et al. (2018), at the test stage, we compute the Batch Normalization statistics for a network on the curve with an additional pass over the data, since these are not collected during training. Except Batch Normalization, other training parameters are kept the same as discussed for Section 3.

Figure 13(a) shows the training loss for models along the line segment and MC curve joining the pair of iterates from SGDR. For the two pairs $\{w_{30}, w_{150}\}$ and $\{w_{70}, w_{150}\}$, we again observe a higher training loss for models on the line segment, suggesting that for this setup too, SGDR finds paths that move over a barrier in the training loss landscape. We further evaluate the intermediate points

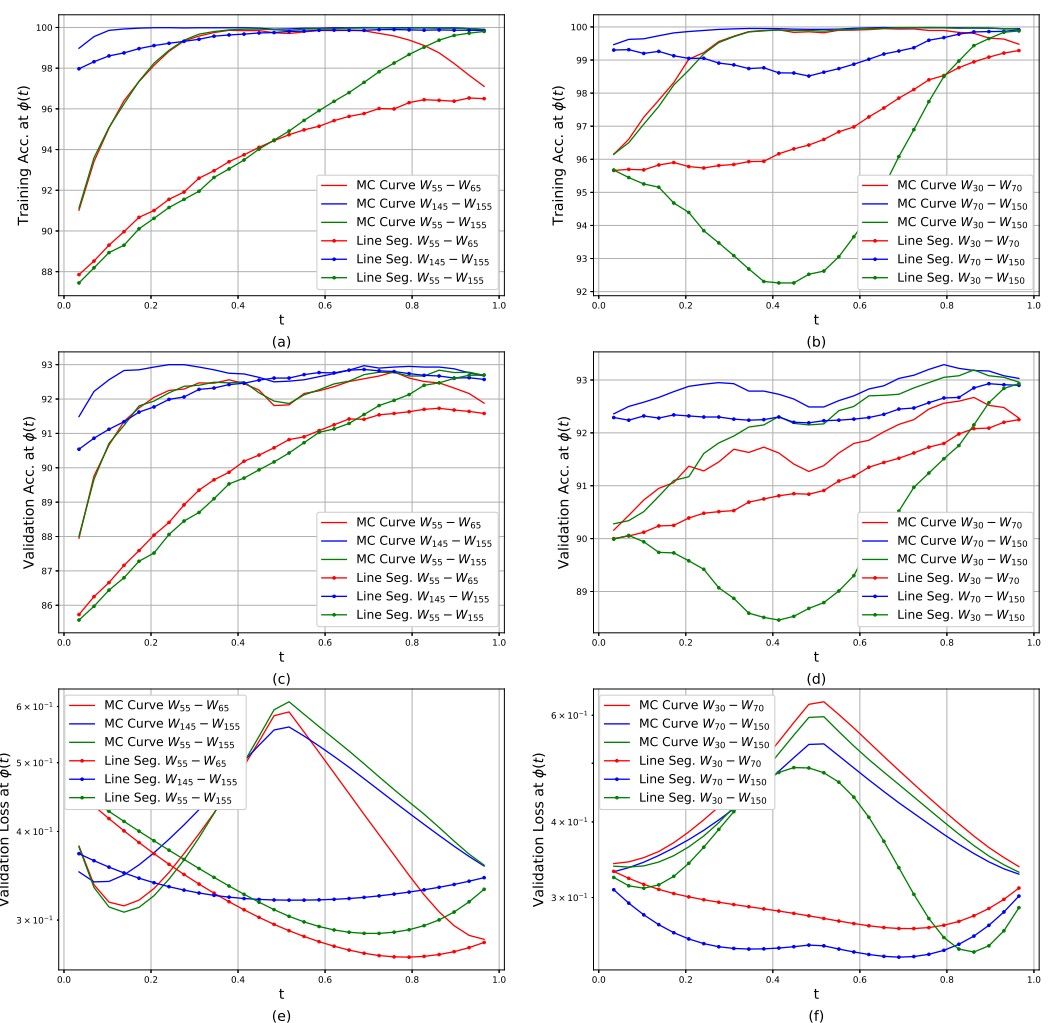

Figure 11: **Left** Column: Connecting iterates from SGD with step-decay learning rate scheme **Right** Column: Connecting iterates from SGDR **Top** Row: Training Accuracy on the curve found through Mode Connectivity (MC Curve) and on the line segment (Line Seg.) joining iterates from SGDR and SGD. **Middle** row: Validation Accuracy on the curve found through Mode Connectivity (MC Curve) and on the line segment (Line Seg.) joining iterates from SGDR and SGD. **Bottom** row Validation Loss on the curve found through Mode Connectivity (MC Curve) and on the line segment (Line Seg.) joining iterates from SGDR and SGD.

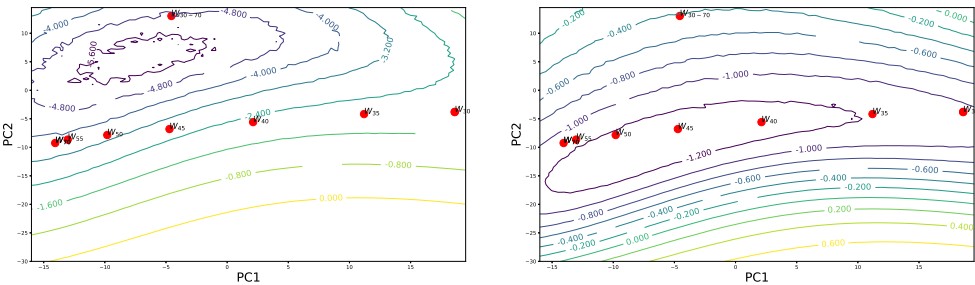

Figure 12: Training Loss (left) and Validation Loss (right) surface (log scale) for points on the plane defined by $\{w_{30}, w_{70}, w_{30-70}\}$ including projections of iterates on this plane

on the hyperplane defined by $\{w_{70}, w_{150}, w_{70-150}\}$ and plot their training and validation loss in Figure 13(b) and (c) respectively. Our previous observations regarding (a) the iterates $w_{70}$ and $w_{150}$ not lying in different basins, (b) the MC-found $\theta$ or $w_{70-150}$ generalizing poorly and (c) averaging of iterates improving generalization hold true here as well.

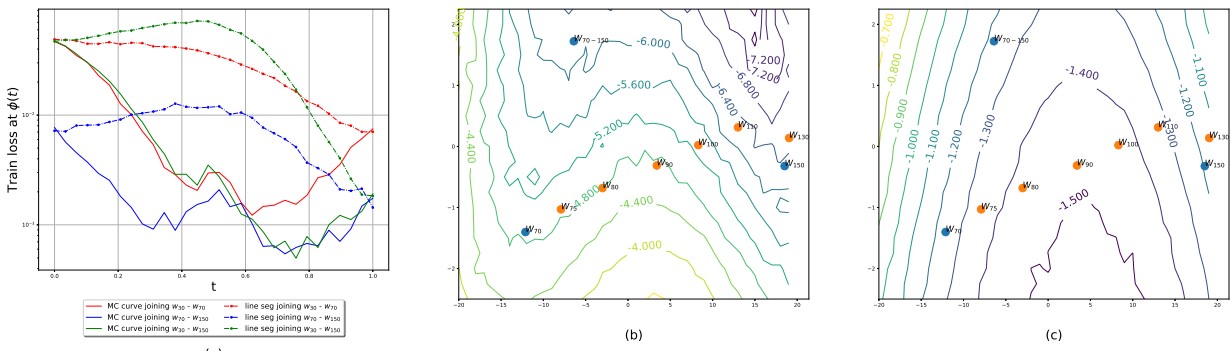

Figure 13: (a) Training loss for points on line segment and MC curve joining the pairs $w_{30} - w_{70}$, $w_{30} - w_{150}$ and $w_{70} - w_{150}$ (b) Training loss surface (log scale) for points on the plane defined by $\{w_{70}, w_{150}, w_{70-150}\}$ including projections of iterates on this plane, (c) Validation Loss Surface (log scale) for points on the plane defined by $\{w_{70}, w_{150}, w_{70-150}\}$ including projections of iterates on this plane

## 9    SGDR CCA HEATMAPS

In Figure 14, we present the CCA similarity plots comparing two pairs of models: epochs 10 and 150, and epochs 150 and 155. The $(i, j)^{th}$ block of the matrix denotes the correlation between the $i^{th}$ layer of the first model and the $j^{th}$ layer of the other. A high correlation implies that the layers learn similar representations and vice versa. We present the former to compare against the typical stepwise or linear decay of SGD, and the latter to demonstrate the immediate effect of restarting on the model. Raghu et al. (2017) showed in their work that for typical SGD training, a CCA similarity plot between a partially and completed trained network reveals that the activations of the shallower layers bears closer resemblance in the two models than the deeper layers. We note that, despite the restart, a similar tendency is seen in SGDR training as well. This again suggests that the restart does not greatly impact the model, both in weights and representations, and especially so in the shallower layers. A comparison of epochs 150 and 155, i.e., before and after a restart also stands as evidence for this hypothesis.

## 10    WARMUP EXPERIMENTS ON RESNET-18 AND RESNET-32

In Figure 4(d), we show that the stability induced by warmup when training with large batches and learning rates can also be obtained by holding the FC stack frozen. This experiment was conducted

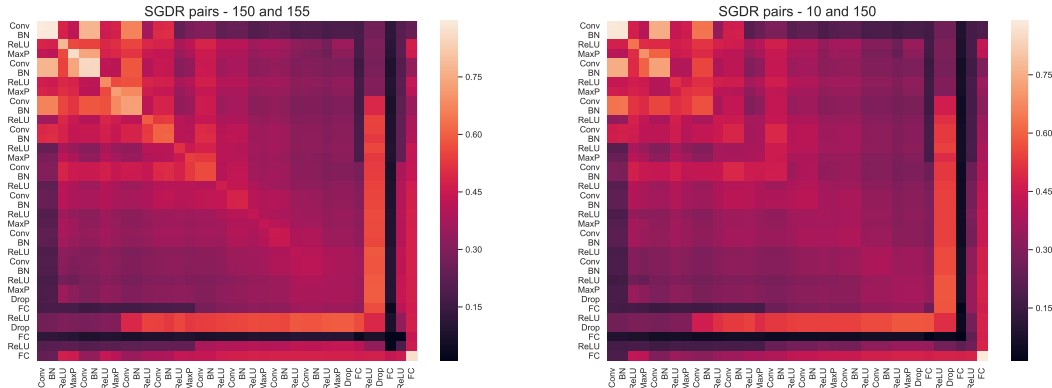

Figure 14: CCA similarity scores between two pairs of models. (a) comparings models at epochs 150 and 155, (b) comparing models at epochs 10 and 150. The $i, j$-th cell in each pane represents the CCA similarity between layer $i$ of $w_a$ (model at epoch a) and layer $j$ of model $w_b$ (model at epoch b).

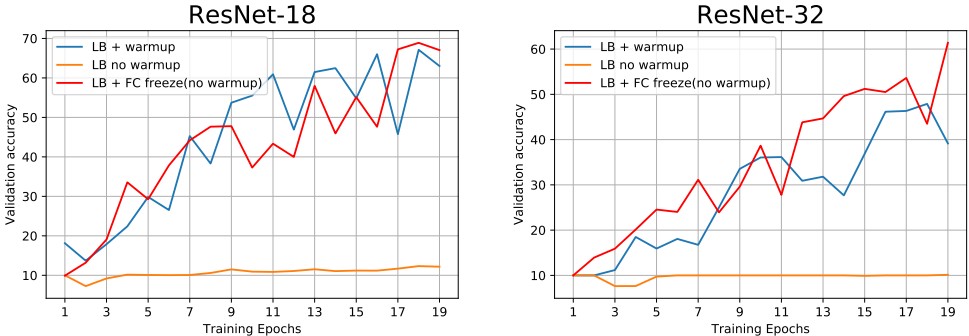

Figure 15: Experiment comparing warmup and FC freezing strategies on ResNet architectures.

on the VGG-11 network (Simonyan & Zisserman, 2014). To demonstrate the generality of our claim, we present additional experiments on two ResNet architectures: 18 and 32. The setup for this experiment is identical to the VGG-11 one with one change: instead of the learning rate being set to 2.5, which is the learning rate for SB (0.05) times the batch size increase (50×), we set it to 5.0 since SB training is better with 0.1. For the warmup case, we linearly increase the learning rate from 0 to 5 again for 20 epochs. Experiments on other configurations yielded similar results. Whether these results remain true also for training larger datasets, such as ImageNet, remains to be shown and is a topic of future research.

