# OpenReview forum: "A Closer Look at Deep Learning Heuristics: Learning rate restarts, Warmup and Distillation"
_ICLR.cc/2019/Conference_

### Official Review · AnonReviewer3 · 2018-10-30
**Interesting empirical study with some flaws**

**Rating:** 6
**Confidence:** 4

**Review:**

This paper empirically explores heuristics commonly used in deep learning: learning rate restarts, warmup and distillation. The authors utilize two recently proposed tools for neural network analysis: mode connectivity (MC) finding a low loss pathway between two given points in the space of DNN parameters and CCA measuring the correlation of  DNN layer activations. Conducting a set of experiments and analyzing the results the authors refine the intuition behind the considered heuristics and dynamics of corresponding training procedures.

Strengths:

+ The authors conduct experiments ensuring robustness of MC framework.
+ In the chosen settings the experimental methodology of the paper sounds reasonable. I find the idea of DNN analysis from both perspectives of weight space and activations important.
+ Paper is well-written and organized clearly. All the used methods and experiments are adequately described.
+ The authors draw connections between obtained results and hypotheses introduced in prior work.

Weaknesses:

- There is a possible flaw in the choice of experimental settings. Authors mention Batch Normalization (BN) among heuristics widely used in deep learning. It is known that properties of both loss surface and activations are different between DNN architectures which include BN layers and those which do not. To emphasize generality of obtained results, it would be beneficial to conduct experiments for both types of DNN architectures as at the moment the majority of the results are presented for VGG architecture which typically does not include BN. Impact of other architecture modifications (e.g. skip connections) might be considered as well.

- I find the significance of the results unclear. Although the particular insights of the learning procedures are revealed there is not enough attention paid to their value for possible improvements of the procedures and their applications. There is only one idea proposed by the authors based on the experimental results – fixing the deeper layers during the warmup phase, but the practical implications of this idea are not discussed.

Other comments:

* The scale used in Figure 3 and similar figures in the appendix is not easily comprehensible. I recommend to comment further on the scale or possibly adjust it.

---

> ### Author Response · Authors · 2018-11-17
> **Response to Reviewer 3**
>
> We thank the reviewer for the feedback.
>
> Our responses to the two weaknesses pointed out in the review:
>
> (1) We agree that our observations can gain a lot more generality by covering variants that do and do not have Batch Normalization (BN). For the loss landscape analysis performed for SGDR iterates, our current implementation of VGG indeed does not involve Batch Normalization. We have added new results in the appendix (Section 8.4 and Fig 13) to cover the case of VGG with BN. The results obtained are qualitatively very similar to the ones for without BN.
> For results related to the FC freezing for large batch training, Section 10 (and Fig 15) in the Appendix includes results for 2 ResNet architectures (containing BN and skip connections).
>
> (2) Regarding practical implication: Thank you for your suggestion. We have added a brief discussion to our paper highlighting the possible practical implications of our work, and specifically: new heuristics that could improve training and fundamental research questions that our results motivate. We hope one valuable takeaway for readers would be our careful investigation of the heuristics and these new questions that our results open up.
>
> Thank you for pointing out the issue with Fig 3, we have modified its caption to aid the reader’s understanding.

---

### Official Review · AnonReviewer1 · 2018-11-01
**Good paper, well presented with thorough experimental work**

**Rating:** 7
**Confidence:** 5

**Review:**

Summary:
This paper uses the recently proposed techniques of mode connectivity and CCA to analyze two different popular heuristics in deep learning:
(1) SGDR (stochastic gradient descent with restarts/cosine annealing of learning rate)
(2) Learning rate warmup
(3) Model distillation

For (1) they visualize 1d and 2d slices of the loss surface either using mode connectivity, or parameter points immediately before restarts to try and understand if the parameters sit in different local minima. For (2), they study the effect of learning rate warmup using CCA, coming to the conclusion that learning rate warmup helps stabilize the fully connected layers. (3) They also study model distillation with CCA, finding out that the higher layers are the most similar to the teacher model.

Clarity: The paper is clearly written, cites lots of relevant work, and describes the experiments in detail.

Originality: This paper seems original, while the techniques used are established, they conduct thorough experiments on phenomena in deep learning that haven't been studied.


Comments on Significance and Quality:
I liked parts (2), (3) of the paper most, as it seemed like conclusions from these parts were fairly clear:

Figures 4, 5 make the effect of warm restarts in the large batch setting on FC layers clear: the restarts help the layers stabilize better. I really liked the experiment in 4(d), where they tested this hypothesis by freezing the fully connected layers for the duration of the warmup.  It was interesting to see that this had no effect on the remainder of the trajectory. This seemed to be a good demonstration and investigation of the effect of warm restarts, and I appreciate the tests on different architectures in the supplementary material. I'd be curious to see if there's some way to further incorporate this into learning rate schedules.

I also liked Figure 6, exploring Model distillation, which showed that the higher layers of the shallower network were the most affected by the teacher network. The authors cite related work which suggests only training higher layers, and I'd be curious to see how only training higher layers affects accuracy.

While I thought the experiments for part (1) SGD with Restarts were thorough, and appreciated Figure 1, which experimentally validated the use of mode connectivity, I felt there was some difficulty in interpreting the results.

Firstly, in Figure 2, the claim is that SGD with Restarts does possibly bridge local minima as the mode connectivity curves increase between the two convergence points. However, we see in both 2(b) and 2(c) that the linear interpolation between both convergence points does *not* increase in loss. In which case is there any reason to believe that the increase of MC in the middle means that SGDR is climbing a basin? How do we know that the linear combination isn't closer to the path followed by SGDR?

For additional comparisons, it would be good to have the linear combination plots for Figure 1 also.

In general, it seems hard to make meaningful conclusions with low dimensional projections of a very high dimensional loss surface. We'd have to know some kind of theoretical property of MC to be able to do so.

Minor Comments

I think the figures in this paper could be much clearer. In Figure 2 for example, the legend blocks some of the main areas of interest of the plot. I would recommend cutting some of the raw learning rate figures and making all figures much bigger.

In figure 4(d), the text describes the process in training steps (200 training steps), but the plot is in epochs -- it would be better if the text and axis were consistent in units.

Conclusion:
Despite my concerns on the first part of this paper, I think the very thorough experiments, clear presentation and the interesting results on learning rate warmups and model distillation merit its acceptance.

---

> ### Author Response · Authors · 2018-11-17
> **Response to Reviewer 1**
>
> We thank you for a thorough and supportive review. Our responses to the reviews are below -
>
> Regarding Fig 1:
> As recommended, we have included the validation accuracy for models on line segment joining the endpoints in figure 1.
>
> Regarding Figure 2 and claims related to SGDR:
> We’re afraid there has been a misunderstanding here. We do not claim, or suggest, that SGDR iterates possibly bridge local minima. In fig 2(c) and 2(d) (in original draft), the dot dash curve corresponds to the line segment joining the two iterates and the solid curve represents the MC curve found between the two. We see a high loss region on the line segment joining SGDR iterates (dot-dash curves) (2c in original draft), while this ‘bump’ is not observed in the case of iterates from SGD (2d in original draft). The MC curves are plotted to indicate the existence of low-loss paths connecting the iterates and highlighting the fact that the SGD iterates ‘jump over the barriers’ (as defined in footnote 3) when there exists a low loss path connecting the two.
>
> Response to minor comments:
> Thank you for pointing out these issues about Fig 2 and Fig 4(d). We have made the recommended changes in our updated manuscript.

---

### Official Review · AnonReviewer2 · 2018-11-02
**Significance of the findings?**

**Rating:** 4
**Confidence:** 4

**Review:**

In this paper, authors propose a set of  control experiments in order to get a better understanding of different deep learning heuristics: stochastic gradient with restart (SGDR),  warmup and distillation. Authors leverage the recently proposed mode connectivity (which fits a simple piecewise linear curve to obtain a low loss path that connect two points in parameter space) and CCA is a way to compute a meaningful correlation of the networks activations. All the experiments are done using a VGG-16 networks on CIFAR10.

For SGDR, authors observe that the solutions found by SGDR or SGD does not appears to be in different basins. While this contradict previous claim, it goes in the same direction than recent works which  have similar observations for the small batch/large batch case [1]. Authors also identify that warmup tends to avoid large change the top-layers at the beginning of training and that you can achieve similar effect than warmup by freezing the top-layer. Finally authors show that most of the benefit of distillation happen by impacting the last deep layers of a network.

While I find all those findings valuable, it is not straightforward to see how they connect to a better understanding of training deep network and how significant they are. In particular,  it is still unclear to me why heuristics such as SGDR is successful in practice or why freezing the top layer of a network improve trainability in a large batch setting?

Doing control experiments in order to better understand the current practice in deep learning is extremely important, however, I don’t think that the paper in its current shape is ready for publication.

[1] Empirical Analysis of the Hessian of Over-Parametrized Neural Networks (Sagun et al., 2017).

---

> ### Public Comment · ~Ilya_Loshchilov1 · 2018-11-06
> **"While this contradict previous claim"**
>
> I would like to intervene and comment "While this contradict previous claim" to avoid possible misunderstandings. The paper does not contradict the original SGDR paper and the authors correctly mention "Loshchilov & Hutter (2016) do not claim to observe any effect related to multi-modality". The paper clarifies that the speculations around SGDR escaping local optima introduced in more recent papers and blog posts don't have enough support ("Huang et al. (2017)’s claims about SGDR converging to and escaping from local minima might be an oversimplification"). These speculations might be confirmed (e.g., I would not be surprised if it happens on some tricky datasets/tasks) but as the paper points out, we are currently lacking enough supporting material to make that claim.
> In line with our observations, the paper notes (see footnote 4) that the cosine annealing part itself makes a major impact and the restarts are primarily to obtain better anytime performance or to benefit from model snapshot ensembling and early stopping. This represents a possible explanation "why heuristics such as SGDR is successful in practice".
> I think that the paper clarifies a few misconceptions around SGDR.

---

> > ### Author Response · Authors · 2018-11-17
> > **Thank you for the clarification**
> >
> > Thank you for your comment and clarifying the contradiction being discussed. We agree with the points you raised, and are glad that you found our work useful in clarifying some aspects of SGDR.

---

> ### Author Response · Authors · 2018-11-17
> **Response to Reviewer 2**
>
> Thank you for your review and helpful comments.
>
> Our observations are indeed suggestive of the connected components at the bottom of the landscape as referred to in Sagun et al. Thank you for pointing us to that work.
>
> Regarding significance:
> The primary goal of our work was to investigate heuristics carefully, and specifically to understand whether hypotheses aimed at explaining them are founded empirically, and also to reveal new insights about how they work. We hope that this is a valuable takeaway for readers in itself to help motivate new techniques and also help answer fundamental questions about loss landscapes and their interaction with training algorithms. While it is true that our work is not conclusive in explaining why and how the heuristics work, we believe that the results are significant at clearing misconceptions and shedding light on a difficult problem, and in turn, raising more interesting questions, such as the ones you mentioned.

---

### Meta-Review · Area_Chair1 · 2018-12-11
**Interesting analysis on the effects of several learning heuristics**

**Confidence:** 4
**Recommendation:** Accept (Poster)

**Metareview:**

The presented method uses mode connectivity to help illustrate the surfaces of parameter space between various selections of models (either through changes of parameters, learning methods, or epochs), and canonical correlation analysis (CCA) to visualize the similarity of model layers across two different selected models.  These analyses are then used to study 3 forms of learning heuristics: stochastic gradient descent with restart (SGDR), warmup, and distillation.

Reviews tend to be leaning toward acceptance.

Pros:
+ R1: Well-written
+ R1: Papers that analyze learning strategies are generally informative to the larger community. These experiments haven't been previously performed.
+ R1: Thorough experiments
+ R3: Results brought into context of prior hypotheses

Cons:
- R3: Batch normalization not studied, but authors have added experiments in response.
- R3 & R2: Practical implications not clear, but authors have added a discussion.